# Extracts from the Leaf of *Couroupita guianensis* (Aubl.): Phytochemical, Toxicological Analysis and Evaluation of Antioxidant and Antimicrobial Activities against Oral Microorganisms

**DOI:** 10.3390/plants12122327

**Published:** 2023-06-15

**Authors:** Marco Aurélio Carmona Augusco, Daniela Abram Sarri, Juliane Farinelli Panontin, Maria Angélica Melo Rodrigues, Rachel de Moura Nunes Fernandes, Juliana Fonseca Moreira da Silva, Claudia Andrea Lima Cardoso, Magale Karine Diel Rambo, Elisandra Scapin

**Affiliations:** 1Postgraduate Program in Environmental Sciences—CIAMB, Federal University of Tocantins, Campus of Palmas, Palmas 77001-090, TO, Brazilpanontin.juliane@mail.uft.edu.br (J.F.P.); magalerambo@mail.uft.edu.br (M.K.D.R.); 2Environmental Engineering Course, Chemistry Laboratory, Block II, Federal University of Tocantins, Campus of Palmas, Palmas 77001-090, TO, Brazil; 3Postgraduate Program in Biodiversity and Biotechnology—BIONORTE, Federal University of Tocantins, Campus of Palmas, Palmas 77001-090, TO, Brazil; 4Medical Course, Laboratory of General and Applied Microbiology, Federal University of Tocantins, Campus of Palmas, Palmas 77001-090, TO, Brazil; julianafmsilva@uft.edu.br; 5Center for Studies in Natural Resources, State University of Mato Grosso do Sul, Dourados 79804-970, MS, Brazil; claudia@uems.br

**Keywords:** abricó de macaco, antioxidant potential, chemistry characterization, dentistry, microbiology

## Abstract

The study of phytotherapy in dentistry holds great relevance because of the scarcity of research conducted on the treatment of oral pathologies, specifically, caries and periodontal disease. Therefore, this research aimed to analyze the chemical composition of extracts from *Couroupita guianensis* Aubl. leaves, evaluate their toxicity, and assess their antioxidant and antimicrobial properties against *Staphylococcus aureus*, *Streptococcus mutans*, and *Candida albicans*. Three extracts were prepared using assisted ultrasound and the Soxhlet apparatus, namely, Crude Ultrasound Extract (CUE), Crude Soxhlet Extract (CSE), and the Ethanol Soxhlet Extract (ESE). Flavonoids, tannins, and saponins were detected in the chemical analysis, while LC-DAD analysis revealed the presence of caffeic acid, sinapic acid, rutin, quercetin, luteolin, kaempferol, and apigenin in all extracts. GC-MS analysis identified stigmasterol and β-sitosterol in the CUE and CSE. The ESE showed higher antioxidant activity (2.98 ± 0.96 and 4.93 ± 0.90) determined by the DPPH• and ABTS•^+^ methods, respectively. In the toxicity evaluation, the CUE at 50 μg/mL and the ESE at 50 μg/mL stimulated the growth of *Allium cepa* roots, while all extracts inhibited root growth at 750 μg/mL. None of the extracts exhibited toxicity against *Artemia salina*. Antibacterial activity was observed in all extracts, particularly against the microorganisms *S. aureus* and *S. mutans*. However, no antifungal activity against *C. albicans* was detected. These results suggest that extracts of *C. guianensis* have therapeutic potential for controlling microorganisms in the oral microbiota.

## 1. Introduction

The Legal Amazon is a region known for its heterogeneous space and abundant natural, social, and cultural resources [1,2,3,4]. Recognizing the value of these resources, the region encourages their conservation by integrating various economic and social interests while prioritizing the preservation of its unique ecosystem. With its exceptional biodiversity, the Legal Amazon is at the forefront of herbal medicine research, offering extensive possibilities for discovering new drugs owing to the unparalleled availability of chemical diversity [5].

Within the scientific community, there is a growing interest in discovering medicinal plant species for dental applications [6,7,8,9]. This interest has fueled research focused on developing products with substantivity, minimal harm to oral tissues [10], reduced bacterial biofilm [11], and unfavorable conditions for the growth of resistant bacteria [12]. As a result, phytotherapy has gained recognition within the dental community for providing natural products that exhibit superior biocompatibility, lower toxicity, and scientifically proven therapeutic activity compared to conventional medicines.

One such plant species native to the Legal Amazon is *Couroupita guianensis* Aubl., commonly known as “abricó de macaco” in Brazil [13]. Various medicinal properties have been attributed to different parts of this plant, including the leaves, flowers, fruits, roots, stems, and seeds. These properties range from the treatment of malaria [14], hypertension, and anti-inflammatory effects [15] to healing properties [16], as well as analgesic and antimicrobial activity [14].

In a study by Ngo et al. [17], four triterpenoids (betulinic acid, oleanolic acid, β-amyrin, and friedelin) were identified in the fruit and leaf of *Couroupita guianensis* Aubl. These compounds were previously reported in the literature to possess antimicrobial properties [18,19,20,21,22,23]. Additionally, *C. guianensis* was found to contain various other compounds, such as flavonoids, saponins, quercetins, alkaloids, and volatile compounds, as documented in previous studies [24].

Despite existing studies in the field of microbiology, the potential of *Couroupita guianensis* Aubl. in controlling oral pathogenic microorganisms remains relatively unexplored. Therefore, the aim of this research was to conduct a comprehensive phytochemical and toxicological analysis of leaf extracts from *Couroupita guianensis* Aubl. Furthermore, the study aimed to evaluate the antioxidant and antimicrobial activities of these extracts, with a specific focus on microorganisms commonly found in the oral cavity.

## 2. Results and Discussion

### 2.1. Chemical Composition

Table 1 presents the results obtained for the phytochemical screening of *C. guianensis* leaf extracts.

Flavonoids, tannins, and saponins were detected in all extracts, indicating their presence in *C. guianensis* leaf extracts. Natural substances such as flavonoids and tannins have demonstrated antioxidant activity [25]. Moreover, tannins have been shown to play a significant role in antimicrobial activity by interfering with bacterial adherence to tooth surfaces, as observed in studies on Gram-positive and Gram-negative bacteria present in oral biofilms [26]. Triterpenoids were identified in the CUE and CSE, while alkaloids were found in the CSE and ESE. Research conducted by Achika et al. [27] highlighted the importance of terpenes and terpenoids in controlling multi-resistant bacteria that are not responsive to conventional antimicrobials.

The presence of flavonoids, tannins, terpenoids, and alkaloids in *C. guianensis* leaf extracts offers a wide range of therapeutic actions, including antiviral, antifungal, antiprotozoal, antioxidant, and anti-inflammatory effects. Additionally, these compounds inhibit cell growth and division, playing a crucial role in preventing the formation of bacterial biofilms on the surfaces [28,29,30,31].

### 2.2. Content of Phenolic Compounds, Tannins, Flavonoids, Flavonols and Evaluation of Antioxidant Activity

Table 2 provides the quantification of phenolic compounds, tannins, flavonoids, and flavonols, along with the results of the antioxidant activity for the leaf extracts of *Couroupita guianensis*.

In the determination of the flavonol and flavonoid contents, among all the analyzed extracts, the CUE presented the highest amounts (100.89 ± 1.05 and 307.21 ± 1.05 mgRE/g, respectively). Regarding the phenolic compounds (CUE = 85.58 ± 0.51 mg GAE/g, CSE = 90.19 ± 0.29 mg GAE/g, ESE = 92.31 ± 0.38 mg GAE/g) and tannins (CUE = 20.96 ± 0.62 mg GAE/g, CSE = 17.69 ± 0.29 mg GAE/g, ESE = 19.62 ± 0.88 for the ESE), the three extracts analyzed did not show a statistically significant difference between them. On the other hand, the extract that presented the highest content of the flavonoids and flavonols was the CUE, obtained by assisted ultrasound. In the DPPH• and ABTS^•+^ radical scavenging assay, the ESE showed greater efficiency in the scavenging capacity of free radicals (IC_50_ of 2.98 ± 0.96 and 4.93 ± 0.90 μg/mL, respectively), obtaining better results than the positive controls. It was observed that, in this case, the antioxidant activity was potentiated in the extract obtained after the sample degreasing process.

Phenolic compounds have been recognized for their antioxidant effects, which can aid in the prevention of various types of cancer and cardiovascular diseases [32]. In the field of endodontics, phenolic acids show promise as healing agents between treatment sessions, facilitating root canal disinfection, preventing inflammation, and promoting the repair of the periapical tissue [33].

The levels of phenolic compounds found in this work were comparable to those reported by Sirisha and Jaishree [29] in the methanolic extract of *C. guianensis* (96.90 mg EAG/g of extract) with demonstrated antioxidant activity. Akther et al. [24] quantified the flavonoid content in the methanolic extract of different parts of *C. guianensis*, revealing that the leaves have the highest flavonoid content, followed by the flower, fruit pulp, fruit peel, and bark stem.

Additionally, Sathishkumar et al. [34] determined the total phenolic content of *C. guianensis* extract (343 ± 0.8 mg EAG/g) and observed a significant DPPH• radical scavenging antioxidant activity (IC_50_ = 37 μg/mL) at a concentration of 100 μg/mL. This finding provided evidence of the relationship between the total phenolic content and the antioxidant activity of the extract. In similar studies, *C. guianensis* leaf extracts were analyzed and also exhibited DPPH• radical scavenging activity (IC_50_ = 18 μg/mL) [35].

Pinheiro et al. [36] conducted a study that highlighted the significant pharmacological potential of *C. guianensis.* They determined the antioxidant activity of different extracts partitioned from the leaf, including ethyl acetate (IC_50_ = 6.38 mg/mL), n-butanol (IC_50_ = 8.19 mg/mL), dichloromethane (IC_50_ = 39.83 mg/mL), hexane (IC_50_ = 64.69 mg/mL), and total ethanolic extract presented (IC_50_ = 19.74 mg/mL).

Furthermore, Raghavendra et al. [37] investigated the free radical scavenging abilities of *C. guianensis* leaf and flower extracts using DPPH• and ABTS•^+^ assays. They observed that the scavenging activity was concentration-dependent for both extracts. The leaf extracts exhibited superior scavenging potential (DPPH•: IC_50_ 19.61 μg/mL; ABTS•^+^: IC_50_ 7.63 μg/mL) compared to the flower extracts (DPPH•: IC_50_ of 257.13 μg/mL; ABTS•^+^: IC_50_ 53.34 μg/mL). These results surpassed the activity of ascorbic acid, used as a standard (DPPH•: IC_50_ 8.89 μg/mL; ABTS•^+^: IC_50_ 3.59 μg/mL).

### 2.3. Analysis by LC-DAD

Table 3 provides a summary of the compounds identified by LC-DAD in the leaf extracts of *C. guianensis* (CUE, CSE, ESE), and Figure 1 displays the corresponding chromatograms obtained from the extracts.

Among the analyzed extracts, several compounds were identified, including caffeic acid, sinapic acid, rutin, quercetin, luteolin, kaempferol, and apigenin. The ESE, obtained after sample degreasing, contained a low amount of apigenin (3.4 ± 0.1 mg/g). Quercetin was the most abundant compound found in all extracts (CUE = 169.8 ± 0.6 mg/g; CSE = 181.4 ± 0.7 mg/g, and ESE = 177.3 ± 0.4 mg/g). These compounds exhibit significant antioxidant activity, with caffeic acid and sinapic acid being particularly noteworthy [38]. Rutin, in addition to its antioxidant activity, has been investigated for its antibacterial and antitumor activities [39].

In a study by Stojkovic et al. [40], rutin was found to inhibit the growth of *Staphylococcus aureus* by up to 100% at a concentration of 1.87 mg/mL and a temperature of 25 °C. Flavonoids such as quercetin, luteolin, kaempferol, and apigenin have also demonstrated antioxidant potential and antimicrobial activity [24]. These identified compounds are significant in the treatment of pathogens that affect the oral microbiota, particularly in inhibiting the formation of bacterial biofilm on the dental surfaces.

No previous literature studies utilizing liquid chromatography with LC-DAD for *C. guianensis* extracts were found.

### 2.4. Analysis by GC-MS

Table 4 present the compounds identified by GC-MS in *C. guianensis* leaf extracts (CUE, CSE, ESE) obtained by ultrasound-assisted and Soxhlet apparatus methods.

The CUE and CSE were found to contain significant amounts of stigmasterol (69.7 ± 0.1 and 85.1 ± 0.2 mg/g, respectively) and β-sitosterol (80.3 ± 0.2 and 91.9 ± 0.3 mg/g, respectively), whereas no compounds were detected in the ESE using this technique. Phytosterols such as β-sitosterol and stigmasterol are known for their antitumor, antifungal, analgesic, and anti-inflammatory properties, which are particularly relevant in the dental field owing to their impact on oral diseases and the formation of bacterial oral microbiota [41,42]. These aspects are central to the focus of our study.

In a related study conducted by Araujo et al. [43] utilizing a similar technique, they identified eight distinct compounds in extracts derived from *C. guianensis* leaves. These compounds encompassed polyphenols and nitrogen-containing alkaloids, among others. Furthermore, in an analysis performed by Venkatraman and Sheba [44], the presence of approximately 30 chemical components was confirmed in the fruit pulp of *C. guianensis*. Notably, the ethanolic extract predominantly contained 2,5-furandione-3-methyl- and 5-hydroxymethylfurfural as its primary constituents.

The analysis of volatile and semi-volatile compounds in mature fruits of *C. guianensis* conducted by Lavanya and John [45] revealed the identification of 50 different compounds, including linalool, benzyl alcohol, terpineol, hexadecanoic acid, and the predominant cis and transfuran oxide linalool.

It is worth noting that the phytosterols identified in our study were not previously reported in other studies analyzing *C. guianensis* leaves using GC-MS.

### 2.5. Toxicity

Table 5 displays the toxicity values of *C. guianensis* leaf extracts against *Allium cepa* (onion) specimens, assessed at various concentrations (50, 250, 750 μg/mL), along with the control group.

When analyzing the results found for the average root length (ARL), significant difference were observed among the studied samples. The 50 μg/mL ESE exhibited the longest root length, while the 750 μg/mL CUE had the shortest. Only in the 50 μg/mL CUE and 50 μg/mL ESE showed a growth stimulus (GS) in the root development. At a concentration of 250 μg/mL, all samples displayed root sizes similar to the control group. However, at a concentration of 750 μg/mL, all extracts inhibited root growth.

The lowest Relative Growth Index (RGI) was observed in the 500 μg/mL CUE. In all extracts, there was an inverse relationship between RGI and extract concentrations, indicating that higher concentration led to lower growth rates (GRs). Extracts with RGI < 0.8 exhibited statistically significant differences compared to other samples (*p* < 0.05).

No studies were found in the literature that specifically used *Allium cepa* for evaluating the effects of *C. guianensis* extracts.

Table 6 presents the IC_50_ values and toxicity rates of *C. guianensis* leaf extracts in the *A. salina* mortality test.

In this study, no toxicity was observed even at the highest concentrations tested. The IC_50_ values of all analyzed extracts (CUE, CSE, and ESE) were above 1000 μg/mL, indicating their classification as non-toxic according to Young et al. [46].

Limited studies utilizing the *A. salina* test to evaluate the toxicity with *C. guianensis* extracts have been reported. Bhuvaneswari et al. [35] evaluated the methanolic extract of *C. guianensis* leaves and found weak activity against *A. salina*, with a 60% elimination rate at a concentration of 6 mg/mL.

In another study, Sivapragasam et al. [47] investigated the effects of *C. guianensis* flower extract on *A. salina* mortality. The general results indicated that the methanolic extract of *C. guianensis* exhibited neither cytotoxic nor genotoxic potential, with an IC_50_ of 1210.65 μg/mL, suggesting its potential development as a therapeutic agent.

### 2.6. Antimicrobial Activity

Table 7 presents the result of the agar diffusion test evaluating the antibacterial activities of *C. guianensis* leaf extracts.

The CUE, CSE, and ESE derived from *C. guianensis* leaves exhibited the formation of inhibition halos against *S. aureus* and *S. mutans* bacteria, surpassing the positive control. However, none of the tested extracts demonstrated fungicidal activity against *C. albicans*. Notably, the inhibition halo size was higher for *S. aureus* compared to *S. mutans*. Among the analyzed extracts, the ESE and CUE displayed the most prominent inhibition halos, outperforming the CSE. A concentration of 200 mg of the extracts effectively inhibited all tested bacteria.

When focusing on *S. aureus* strains, both the CUE (200 mg) and ESE (200 mg and 100 mg) exhibited larger inhibition halos than the positive control, chlorhexidine, and the difference was statistically significant. However, the analyzed extracts were statistically inferior to chlorhexidine when tested against *S. mutans* strains. This aligned with previous research by Singh et al. [28], which also reported the lack of action of *C. guianensis* leaf extract against *C. albicans.*

Akther, Khan, and Hemalatha [24] analyzed the antibacterial activity of flavonoids extracted from different parts of *C. guianensis* and observed inhibition of pathogenic bacteria, supporting the findings of this study and highlighting the antibacterial action of flavonoids.

Raghavendra et al. [37] evaluated the antimicrobial activity of hydroethanolic extracts derived from *C. guianensis* leaves and flowers against Gram-positive and Gram-negative bacteria. Their study confirmed significant antibacterial activity, particularly in the flower extract, reinforcing the results obtained in this study concerning Gram-negative bacteria.

Additionally, Sheba et al. [48] described the antibacterial activity of *C. guianensis* flower extracts against multidrug-resistant strains, including methicillin-susceptible *Staphylococcus aureus* ATCC 29213, methicillin-resistant *S. aureus* (wild type), and methicillin-resistant *S. aureus* (MRSA) ATCC BAA-1026, which supported the findings of this study. On the other hand, Lavanya and John [45] demonstrated the potential of *C. guianensis* leaf extracts against human pathogenic fungi, such as *Candida albicans*, *Cryptococcus* sp., *Microsporum canis*, and *Tri-chophyton rubrum*, which was not observed in the current study.

There are no existing reports in the literature regarding the inhibition of Streptococcus mutans by *C. guianensis* plant extract, making this finding particularly significant for the field of dentistry. The presence of flavonoids, tannins, and saponins detected in the CUE, CSE, and ESE of the *C. guianensis* suggests that they can serve as alternative sources to inhibit bacterial infections.

### 2.7. Determination of the Minimum Inhibitory Concentration (MIC)

For determination the MIC, only the microorganisms *S. aureus* and *S. mutans* were included in the study owing to their demonstrated inhibition halo in the well diffusion test. The CUE and ESE exhibited effectiveness against the *S. aureus* and *S. mutans* at the lowest concentration tested (0.781 mg/mL), indicating bactericidal activity.

In a study conducted by Singh et al. [28], lower MIC values were reported for *S. aureus*, with 25 mg/mL for the ethanolic extract, 50 mg/mL for the methanolic extract, and 100 mg/mL for the chloroform extract.

Furthermore, according to Sheba et al. [48], the *C. guianensis* root extract demonstrated activity against *Propionibacterium acnes* and *Staphylococcus epidermis*, with a minimum MIC values of 0.675 mg/mL and 2 mg/mL, respectively, highlighting its antibacterial potential.

This finding holds particular significance in the field of dentistry, as it demonstrated inhibition of the growth of *S. aureus* and *S. mutans*, which are the primary microorganisms responsible for bacterial plaque formation, as well as the development of caries and periodontal disease.

## 3. Conclusions

The analyzed extracts of *C. guianensis* demonstrated the presence of flavonoids, tannins, and saponins. Among the extracts, the CUE, obtained using ultrasound-assisted methods, exhibited the highest content of flavonols and total flavonoids. Conversely, the ESE, obtained through Soxhlet extraction after degreasing the sample, displayed the most robust antioxidant activity. LC-DAD chromatography identified caffeic acid, sinapic acid, rutin, quercetin, luteolin, kaempferol, and apigenin. These substances are known to possess antioxidant and antimicrobial properties, aligning with the observed results. Additionally, the GC-MS analysis of the CUE and CSE revealed the presence of stigmasterol and β-sitosterol.

Regarding toxicity assessed against *Allium cepa*, only the CUE (50 μg/mL) and ESE (50 μg/mL) stimulated root growth. Furthermore, the extracts exhibited no toxicity toward *Artemia salina* nauplii at the tested concentrations. Antimicrobial activity was observed against *S. aureus* and *S. mutans*, while no antifungal activity was detected against *C. albicans*.

These findings underscored the potential of *C. guianensis* as a valuable source of compounds with significant therapeutic implications, particularly in oral health research. The promising results obtained in this study lay a foundation for further exploration of potential pharmacological applications.

## 4. Experimental Section

### 4.1. Plant Material

The plant material was precisely located at coordinates 10°11′14″ S and 48°19′56″ W. Subsequently, the collected leaves were carefully identified and listed before being added to the herbarium collection of HUTO (Universidade Estadual do Tocantins—UNITINS). To facilitate future reference and research, these leaves were assigned the specific accession number HTO 8057 within the herbarium collection.

In compliance with regulatory requirements, the project involving the collection of *C. guianensis* leaves was duly registered in the National System for the Management of Genetic Heritage and Associated Traditional Knowledge (SISGEN). The registration code A9D18D3 serves as an identification for the project within the system, ensuring adherence to legal protocols and guidelines.

### 4.2. Preparation of Extracts

Following the collection of plant material, a series of steps were undertaken to prepare the samples for extraction. First, the collected plant material was carefully sanitized and subsequently subjected to a drying process in an oven set at 50 °C for a duration of 48 h. Once dried, the material was finely ground using a Willye-type knife mill (Fortinox STAR FT50 brand), and the resulting powder was stored in amber glass bottles to maintain its integrity.

For the extraction process, three different extracts were prepared utilizing two distinct methods: Soxhlet extraction and ultrasound bath extraction. In the ultrasound bath extraction, 5 g of the previously prepared leaf powder was combined with 80 mL of 70% ethanol in an ultrasound bath (USC1600, operating at a frequency of 40 kHz and 135 W). The mixture underwent five cycles of one hour each, at room temperature, resulting in the formation of the Crude Ultrasound Extract (CUE). On the other hand, the Soxhlet extraction method, based on the procedure described by Soares et al. [49] with modifications, involved the use of 5 g of leaf powder and 200 mL of 70% ethanol. The mixture was heated to boiling temperature and maintained for a period of 6 h. This process yielded the Crude Soxhlet Extract (CSE). Similarly, the Ethanol Soxhlet Extract (ESE) was obtained through the same procedure, with the addition of a preliminary degreasing step using hexane. The sample was then dried at room temperature for 24 h before the extraction process was carried out using 70% ethanol.

To remove the solvents, a rotary evaporator (FISATOM) operating at −600 mmHg and 45 °C was employed. Subsequently, the extracts were frozen at −70 °C, followed by lyophilization using a benchtop lyophilizer (LIOTOP L101). The resulting lyophilized extracts were carefully stored in airtight flasks to ensure their preservation for further analysis.

### 4.3. Phytochemical Screening

Phytochemical screening was conducted to identify the main groups of secondary metabolites present in the extracts. The screening involved qualitative tests that relied on specific chemical reactions, resulting in color changes or precipitation. The following groups of secondary metabolites were targeted for detection: flavonoids, tannins, phytosterols, triterpenoids, quinones, saponins, and alkaloids.

To detect flavonoids, the Shinoda Reaction and the sodium hydroxide (5%) test were employed [50,51]. Tannins were identified using four different tests: the ferric chloride 10% test, the lead acetate 10% test, the copper acetate 5% test, and the lead acetate 10% and acetic acid 10% test [51,52]. Phytosterols/triterpenoids were detected using the Liebermann–Burchard test [50,51]. Quinones were identified using chloroform and sodium hydroxide [50,51]. Saponins were detected through the foam test [50,52]. Lastly, alkaloids were identified using Dragendorff’s reagent [51,52].

These qualitative tests were employed to ascertain the presence or absence of the respective secondary metabolite groups in the extracts.

### 4.4. Determination of the Content of Phenolic Compounds

The quantification of total phenolic compounds was carried out using the Folin–Ciocalteu method, with slight modifications based on the procedure described by Amorim et al. [53]. Gallic acid was used as the standard for calibration. For each extract (CUE, CSE, and ESE) or standard (gallic acid, 2–100 μg/mL), 0.2 mL of a methanolic solution (1 mg/mL) was mixed with 0.5 mL of Folin–Ciocalteu reagent (10%). The mixture was then combined with 1 mL of sodium carbonate (75%, *w*/*v*) and 8.3 mL of distilled water, gently stirred, and kept in the dark for 30 min. The absorbance of the resulting solution was measured at 760 nm using a UV-visible spectrophotometer (GLOBAL TRADE TECHNOLOGY, GTA-96).

To determine the total phenolic content, the absorbance values of the samples were interpolated against a calibration curve constructed with various concentrations of gallic acid (y = 0.0052x + 0.038, R^2^ = 0.9915). The results were expressed as milligrams of gallic acid equivalent (GAE) per gram of lyophilized extract. The assay was performed in triplicate on each extract to ensure the accuracy and reliability of the measurements.

### 4.5. Determination of Tannin Content

The tannin content in the CUE, CSE, and ESE was quantified using a modified version of the Folin–Ciocalteu Method combined with casein precipitation, following the procedure described by Amorim et al. [54]. To determine the tannin content, 1 mL of each extract (1 mg/mL) was mixed with 0.1 g of casein and 5 mL of distilled water. The mixture was vigorously stirred until homogenized. After allowing it to stand at room temperature without agitation for 3 h, the solution was then centrifuged at 1358 rpm for 10 min at 10 °C. The non-tannin phenolic constituents present in the supernatant were quantified using the same method as the total phenolic content determination. The amount of tannins was calculated as the difference between the total phenolic content and the non-tannin phenolic content in each extract.

The total tannin content was expressed as milligrams of gallic acid equivalent (GAE) per gram of *C. guianensis* extract. The experiment was conducted in triplicate for each sample to ensure the accuracy and reliability of the results.

### 4.6. Determination of Flavonoid Content

The total flavonoid content in the CUE, CSE, and ESE was determined following the methodology described by Soares et al. [49]. In this method, 0.5 mL of each extract (1 mg/mL) or the rutin standard solution (at concentrations ranging from 10 to 400 μg/mL) was mixed with an aqueous solution of acetic acid (0.5 mL, 60%), methanolic pyridine solution (2 mL, 20%), aluminum chloride (1 mL, 5%), and 6 mL of distilled water. For the blank, methanol was used instead of aluminum chloride. The reaction complex and the blank were carefully shaken and then allowed to stand for 30 min, protected from light. The absorbance of the samples was measured at 420 nm using a spectrophotometer. The total flavonoid content was determined by interpolating the absorbance values of the samples against a calibration curve constructed with the rutin standard. The equation for the calibration curve was y = 0.0019x + 0.0047, with an R^2^ value of 0.9984. The total flavonoid content was expressed in milligrams of rutin equivalent (RE) per gram of lyophilized extract. The analysis was performed in triplicate for each sample to ensure the accuracy and reproducibility of the results.

### 4.7. Determination of Flavonols

The total flavonol content in the *C. guianensis* extracts was measured using the method described by Miliauskas, Venskutonis, and Van Beek [55]. In this method, 0.5 mL of each extract (1 mg/mL) was mixed with 0.5 mL of aluminum chloride solution (20 mg/mL). Then, 1.5 mL of sodium acetate solution (50 mg/mL) was added to the mixture. The resulting solution was incubated at room temperature for 2.5 h. After the incubation period, the absorbance of the samples was measured at 440 nm using a spectrophotometer, calculating the decrease in absorbance of the samples compared to a calibration curve constructed with different concentrations of rutin in methanol (ranging from 2 to 400 μg/mL). The equation for the calibration curve was y = 0.0019x + 0.0047, with an R^2^ value of 0.9984. The results were expressed as milligrams of rutin equivalent per gram of lyophilized extract of *C. guianensis* (mg RE/g). The experiment was performed in triplicate for each sample to ensure the reliability and reproducibility of the measurements.

### 4.8. Assessment of Antioxidant Activity

The antioxidant capacity of the extracts was evaluated using two methods: the DPPH• (2,2-diphenyl-1-picrylhydrazyl) radical scavenging method and the ABTS•^+^ (2,2′-azinobis (3-ethylbenzothiazoline-6-sulfonic acid)) radical cation decolorization assay.

To measure the antioxidant capacity using the DPPH• method, the procedure described by Peixoto-Sobrinho et al. [56] was followed, with rutin used as the positive control. In triplicate, 0.5 mL of various concentrations of the extracts or standards (ranging from 10 to 200 μg/mL) were added to a methanolic solution of DPPH• (3 mL, 40 μg/mL). A blank was prepared by replacing DPPH• with methanol in the reaction mixture. The reaction mixture and the blank were stirred and protected from light for 30 min. After incubation, the absorbances were measured at 517 nm using a spectrophotometer that was calibrated with methanol. The absorbance of the 40 μg/mL DPPH solution was also measured and served as the negative control. The antioxidant activity (AA) was expressed as the percentage of inhibition and calculated using the following equation:$$AA\left(\%\right)=\frac{ [Ac-\left(Aa-Ab\right)]}{Ac}\times 100$$
where AA (%) is the percentage of antioxidant activity; Ac, the absorbance of the negative control; Aa, the absorbance of the sample; Ab, the absorbance of the blank.

The IC_50_ value represents the concentration of the sample required to reduce the absorbance by 50% at 517 nm and is expressed in μg/mL.

To assess the antioxidant capacity using the ABTS•^+^ method, the protocol described by Chen et al. [57] and updated by Rabêlo et al. [58] was followed. The ABTS•^+^ solution was prepared by combining 7 mM ABTS (5 mL) with 2.45 mM potassium persulfate (88 μL) and allowing it to incubate at room temperature in the dark for 16 h. Subsequently, the solution was diluted with 80% ethanol to achieve an absorbance of 0.700 ± 0.020 at 734 nm. A mixture of 2.7 mL of the ABTS•^+^ solution and 0.3 mL of the test samples was prepared. The reaction mixture was then left to stand at 30 °C for 30 min, followed by the measurement of absorbance at 734 nm using a spectrophotometer. The percentage of inhibition AA (%) was calculated as previously described. The IC_50_ value was also calculated to measure the concentration of a sample needed to decrease the absorbance by 50%. IC_50_ was expressed in μg/mL.

### 4.9. Analysis by Liquid Chromatography with Diode Array Detection (LC-DAD)

*C. guianensis* leaf extracts were prepared by solubilizing them in a water:methanol mixture (7:3 *v*:*v*) and subjected to analysis using LC-DAD (Liquid Chromatography with Diode Array Detection). The LC system consisted of an LC-6AD Shimadzu chromatograph (Shimadzu, Kyoto, Japan) equipped with a photodiode array detector (PDA) capable of detecting wavelengths ranging from λ = 200 to 800 nm. A ThermoElectron Corporation ODS HYPERSIL column (C-18, 150 mm long × 4.6 mm diameter, Thermo Electron

Corporation, Waltham, USA) was employed in the analysis. The injection flow rate was set at 1 mL/min, and a volume of 10 μL was injected for each analysis. The chromatographic procedure followed the methodology described by Cardoso et al. [59] and all analyses were conducted at a temperature of 25 °C. The mobile phase consisted of two eluents: eluent A, a binary mixture of water with 6% acetic acid and 2 mM sodium acetate, and eluent B, composed of acetonitrile. A gradient elution profile was employed as follows: 0 min with 5% B, 20 min with 15% B, 30 min with 60% B, and 40 min with 100% B.

To identify the compounds present in the samples, standards were used as references, and their absorption spectra and retention times were compared using a DAD scanning detector within the spectral range of 200–800 nm. Quantification of the compounds was performed by constructing calibration curves using linear regression with LC. The linearity of the standards was evaluated within 10 concentration ranges, yielding high coefficients of determination (R^2^). Specifically, caffeic acid, ellagic acid, sinapic acid, vanillic acid, ferulic acid, and gallic acid exhibited an R^2^ value of 0.9994, while rutin, luteolin, apigenin, naringin, kaempferol, and quercetin had an R^2^ value of 0.9996. All standards were obtained from Sigma (Sigma, ≥ 98%, St. Louis, MO, USA). 

### 4.10. Analysis by Gas Chromatography Coupled to Mass Spectrometry (GC-MS)

To prepare the extract for GC-MS analysis, 100 mg of the sample was solubilized in 2 mL of water using ultrasound for 1 min. Then, 2 mL of hexane was added, and the mixture was subjected to ultrasound for 2 min. After allowing the phases to separate, the hexane fraction was separated from the aqueous fraction. The process was repeated by adding 2 mL of hexane to the aqueous fraction. The resulting hexane fractions were dried and suspended in 1000 mL of hexane. Prior to GC-MS analysis, the solution was filtered through a 0.45 μm Ultrafilter. GC-MS analysis was conducted using a GC-2010 Plus system (Shimadzu, Kyoto, Japan) equipped with a mass spectrometry detector (GC-MS Ultra 2010). A silica capillary column with a 15 m length, 0.2 mm diameter, and 0.2 μm thick film coated with LM-5 (5% phenyl methylpolysiloxane) was utilized.

The analysis was performed under the following conditions: helium was used as the carrier gas at a purity of 99.999% with a flow rate of 1 mL/min. The injection volume was set to 1 μL, and the split ratio was 1:20. The initial oven temperature was set at 150 °C, followed by heating at a rate of 15 °C/min from 150 °C to 280 °C, and a hold at 280 °C for 15 min. The injector and detector temperatures were both set to 280 °C. In the MS scan, an electron impact ionization voltage of 70 eV was applied, and the mass range was set from 45 to 600 nm m/z with a scan interval of 0.3 s.

Compound concentrations were determined by external calibration, and the linearity of the standards was assessed within 5 concentration ranges. The mean standard errors for the peak areas of replicate injections (*n* = 5) were found to be less than 2%, indicating good repeatability of the calibration curve. The coefficient of determination (R^2^) was 0.9996 for stigmasterol and β-sitosterol. All standards used in the analysis were purchased from Sigma (Sigma, ≥ 98%, St. Louis, MO, USA).

### 4.11. Toxicity Evaluation

The evaluation of the toxicity of the extracts was determined by the *Allium cepa* and *Artemia salina* methods.

Toxicity analysis on *Allium cepa* was carried out according to the methodology of Meneguetti et al. [60] with modifications. The specimens of *Allium cepa* were acquired in the local market, with standardization of size, origin, not germinated and healthy aspect. After removing the peel, the bulbs were submerged in 50 mL of mineral water for 48 h at 25 ºC to identify healthy onions. After this period, the onions had their roots trimmed and then submerged in 50 mL of aqueous extracts of *C. guianensis* leaves under different concentrations (50, 250, 750 μg/mL), and the control group was submerged in 50 mL of mineral water at 25 °C for five days. At the end of the period, the number of germinated roots of each onion was counted and the three largest were measured with a digital caliper. Analyses were performed in triplicate.

The experimental design was completely randomized in a factorial scheme, an arrangement (4 × 3) with 9 treatments: the control, three concentrations of each extract and three replications per treatment. The Relative Growth Index (RGI) was determined through the root growth of the control and the extracts according to Young et al. [46] obtained by the equation:RGI = RLS/RLC
where RLS is the root length sample and RLC is the root length control. The effect of the extract in relation to the control was determined as a function of the RGI, which is subdivided into 3 categories: RGI < 0.8: Growth inhibition (GI); 0.8 ≤ RGI ≤ 1.2: Same effect as control (SCE) and RGI > 1.2: Growth stimulus (GS).

In another toxicity test, the biological activity of *C. guianensis* extracts was evaluated using *Artemia salina* L. nauplii, with the Mean Lethal Concentration (*IC*_50_) serving as the parameter. The methodology employed was based on McLaughlin et al. [61]. A total of 25 mg of *A. salina* eggs were incubated in a container filled with artificial sea water composed of NaCl (23 g/L), MgCl_2_·6H_2_O (11 g/L), Na_2_SO_4_ (4 g/L), CaCl_2_·2H_2_O (1.3 g/L), and KCl (0.7 g/L). The incubation temperature ranged from 20 to 30 °C, and the pH was maintained between 8.0 and 9.0, adjusted using Na_2_CO_3_. After 24 h, viable nauplii exhibiting motor activity were selected for the test.

The assays were conducted in triplicate to establish a dose–response relationship. The control group consisted of artificial sea water only, while the test groups comprised artificial sea water containing the CUE, CSE and ESE of *C. guianensis* at concentrations of 50, 500, 1000, and 5000 μg/mL. Each tube, including the control group, contained 10 nauplii of *A. salina* and was filled to a final volume of 5 mL with artificial sea water. The tubes were then incubated in the dark for 24 h. Following the incubation period, immobile nauplii were counted and the IC_50_ was calculated using the Probits statistical method. The classification of the extract followed the criteria established by Nguta et al. [62] with values of IC_50_ < 100 μg/mL indicating high toxicity; 100 ≤ IC_50_ ≤ 500 μg/mL indicating moderate toxicity; 500 < IC_50_ ≤ 1000 μg/mL indicating weak toxicity; IC_50_ > 1000 μg/mL indicating non-toxic.

### 4.12. Antimicrobial Activity Evaluation

The bioassays utilized ATCC (American Type Culture Collection) reference strains obtained from the collection of the Faculty of Dentistry of the University of São Paulo (USP-Bauru). The strains selected for the study were *Staphylococcus aureus* (ATCC 6538), *Streptococcus mutans* (ATCC 25175), and the fungus *Candida albicans* (ATCC 90028).

To maintain viability, the strains were stored in Brain Heart Infusion (BHI) broth at a freezing of −70 °C. Prior to experimentation, the strains were reactivated by subculturing on nutrient agar and incubating at 35 °C for 24 h. Following reactivation, the strains were stored in a refrigerator at 2–8 °C until the time of preparation and standardization for the testing.

### 4.13. Antimicrobial Test by the Agar Diffusion Method (Well)

The antimicrobial activity was assessed using the well diffusion method, with the experiment conducted in triplicate and repeated three times. Petri dishes (140 × 15 mm) containing 50 mL of Mueller–Hinton agar (AMH) medium for bacteria and Sabouraud agar medium for yeast were prepared. The extracts (CUE, CSE and ESE) were diluted in a mixture with 10% dimethylsulfoxide (DMSO) at concentrations of 200, 100, and 50 mg/mL. Inoculum solutions were prepared using isolated strains, diluted in 0.85% saline solution to achieve a turbidity of 0.5 on the MacFarland scale. This resulted in approximately 1.5 × 10^8^ colony-forming units per milliliter (CFU/mL) for bacteria and 2.0 × 10^6^ CFU mL of yeast. For the positive control, chlorhexidine (Periogard^®^ 0.12%, Colgate, São Bernardo do Campo, São Paulo, Brazil) was used at concentrations ranging from 1.1 to 0.00234 mg/mL for bacteria, while nystatin (100,000 IU/mL) was used for the fungus. The negative control consisted of a 10% DMSO solution, following the protocol by Oliveira et al. [63] with adaptations.

Using a sterile swab, the inoculum solutions were evenly spread on the surface of the culture medium plates. Then, wells were created in the agar using a sterile plastic straw with a diameter of 5 mm. Each well was filled with 50 μL of each extract, diluted in 10% DMSO, along with the positive and negative controls [64]. Following an incubation period of 24 h for bacteria and 48 h for the fungus at 37 °C, the results were analyzed by measuring the diameters of the inhibition zones surrounding the wells. A digital caliper model Starret 799 was used for accurate measurements.

### 4.14. Determination of the Minimum Inhibitory Concentration (MIC)

The tests were conducted in triplicate with three replicates using a sterilized 96-well microplate (ELISA plate). Each microplate was dedicated to the extract test [65], which involved the microorganisms *Staphylococcus aureus*, *Streptococcus mutans*, and the fungus *Candida albicans*.

To initiate the test, 100 μL of Mueller–Hinton broth (M-H broth) was added to each well. Serial dilutions were prepared for the tested extracts, ensuring that each subsequent concentration was half of the previous one. The concentrations of the extracts tested in the experiment ranged from 100 to 0.78125 mg/mL (100, 50, 25, 12.5, 6.25, 3.125, 1.5625, and 0.78125 mg/mL). The positive control (C+) consisted of 0.12% chlorhexidine while the negative control (C−) was 10% DMSO. The culture medium (CM) control consisted of only 100 μL of M-H Broth. For the microbial growth control (C), 5 μL of bacterial suspension, 107 UFC/mL bacterial suspension, was added to the culture medium. The plates were incubated at 37 °C for 24 h [63].

After the incubation period, 30 μL of sterile 0.03% (*w*/*v*) resazurin solution was added to each well and then the plates were reincubated for 2 to 4 h or until the dye changed color. The presence of blue color represented the absence of growth (growth inhibition) and pink color represented the presence of bacterial growth (no growth inhibition) [66]. To evaluate the bacteriostatic/bactericidal activity of the extracts, an aliquot (100 μL) of the lowest inhibitory concentration was inoculated on Mueller–Hinton agar and incubated at 37 °C for 24 h. The presence of bacterial growth indicated a bacteriostatic effect, while the absence of microbial growth indicated a bactericidal effect.

### 4.15. Statistical Analysis

The experiments were conducted in triplicate, and the results were expressed as mean ± standard deviation. Statistical analysis was performed using the SISVAR program version 5.6 [67] and GraphPad Prism 8. Analysis of variance (ANOVA) was employed to compare the mean values obtained in the analyses. A significance level of *p* < 0.05 was considered statistically significant, as determined by the Tukey test.

## Figures and Tables

**Figure 1 plants-12-02327-f001:**
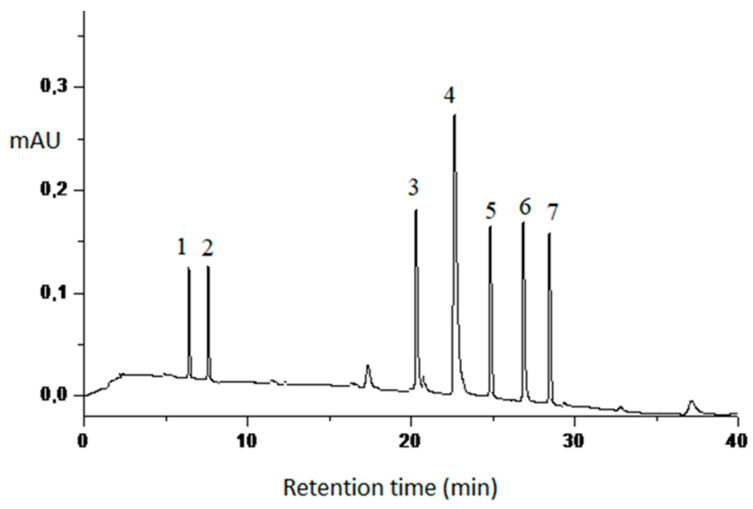
Representative chromatogram (LC-DAD) of *C. guianensis* leaf extracts. Peak 1: caffeic acid; peak 2: sinapic acid; peak 3: rutin; peak 4: quercetin; peak 5: luteolin; peak 6: kaempferol; peak 7: apigenin.

**Table 1 plants-12-02327-t001:** Phytochemical screening of *C. guianensis* leaf extracts obtained by ultrasound and Soxhlet.

Chemical Group	Extract		
	CUE	CSE	ESE
Flavonoids	+	+	+
Tannins	+	+	+
Phytosterols	−	−	−
Triterpenoids	+	+	+
Quinones	−	−	−
Saponins	+	+	+
Alkaloids	−	+	+

(+) presence; (−) absence. CUE—Crude Ultrasound Extract; CSE—Crude Soxhlet Extract; ESE—Ethanol Soxhlet Extract.

**Table 2 plants-12-02327-t002:** Contents of phenolic compounds, tannins, flavonoids, flavonols and antioxidant activity (DPPH• and ABTS•^+^) of extracts of leaves of *Couroupita guianensis* obtained by ultrasound and Soxhlet and positive controls rutin and gallic acid.

	Extract			Control	
	CUE Mean ± SD	CSE Mean ± SD	ESE Mean ± SD	Gallic Acid Mean ± SD	Rutin Mean ± SD
Phenolic compounds (mg GAE/g)	85.58 ± 0.51 ^a^	90.19 ± 0.29 ^a^	92.31 ± 0.38 ^a^	-	-
Tannins (mg GAE/g)	20.96 ± 0.62 ^a^	17.69 ± 0.29 ^a^	19.62 ± 0.88 ^a^	-	-
Flavonoids (mg RE/g)	307.21 ± 1.05 ^a^	101.07 ± 1.99 ^b^	65.63 ± 0.53 ^c^	-	
Flavonols (mg RE/g)	100.89 ± 1.05 ^a^	88.61 ± 0.80 ^b^	56.33 ± 0.80 ^c^	-	-
DPPH• IC_50_ (μg/mL)	59.51 ± 0.26 ^c^	31.13 ± 0.55 ^b^	2.98 ± 0.96 ^a^	-	11.92 ± 0.47
ABTS•^+^ IC_50_ (μg/mL)	30.32 ± 1.60 ^c^	15.74 ± 1.45 ^b^	4.93 ± 0.90 ^a^	6.75 ± 0.01	-

Values followed by the same letter indicate significant similarities in the same column (*p* < 0.05, ANOVA followed by Tukey’s test); values represent the mean followed by the standard deviation (Mean ± SD). SD: Standard deviation. CUE—Crude Ultrasound Extract; CSE—Crude Soxhlet Extract; ESE—Ethanol Soxhlet Extract.

**Table 3 plants-12-02327-t003:** Identification of chemical composition by LC-DAD of *C. guianensis* leaf extracts (mg/g ± DP) obtained by ultrasound and Soxhlet.

Compound	Concentration (mg/g)		
	CUE Mean ± SD	CSE Mean ± SD	ESE Mean ± SD
Caffeic acid	37.6 ± 0.1 ^a^	39.4 ± 0.2 ^a^	38.5 ± 0.2 ^a^
Sinapic acid	37.1 ± 0.1 ^a^	38.6 ± 0.1 ^a^	38.1 ± 0.1 ^a^
Rutin	124.1 ± 0.4 ^a^	138.1 ± 0.6 ^a^	129.2 ± 0.5 ^a^
Quercetin	169.8 ± 0.6 ^a^	181.4 ± 0.7 ^a^	177.3 ± 0.4 ^a^
Luteolin	100.1± 0.2 ^a^	103.2 ± 0.4 ^a^	101.8 ± 0.3 ^a^
Kaempferol	94.8 ± 0.2 ^a^	97.4 ± 0.3 ^a^	96.3 ± 0.2 ^a^
Apigenin	75.2 ± 0.1 ^a^	78.8 ± 0.2 ^a^	3.4 ± 0.1 ^b^

Values followed by the same letter indicate significant similarities in the same column (*p* < 0.05, ANOVA followed by Tukey’s test); values represent the mean followed by the standard deviation (Mean ± SD). SD: Standard deviation. CUE—Crude Ultrasound Extract; CSE—Crude Soxhlet Extract; ESE—Ethanol Soxhlet Extract.

**Table 4 plants-12-02327-t004:** Identification of chemical composition by GC-MS of extracts from leaves of *C. guianensis* (mg/g± DP) obtained by ultrasound and Soxhlet.

Compound	Concentration (mg/g)		
	CUE Mean ± SD	CSE Mean ± SD	ESE Mean ± SD
Stigmasterol	69.7 ± 0.1 ^b^	85.1 ± 0.2 ^a^	- *
β-sitosterol	80.3 ± 0.2 ^a^	91.9 ± 0.3 ^a^	- *

Values followed by the same letter indicate significant similarities in the same column (*p* < 0.05, ANOVA followed by Tukey’s test); Values represent the mean followed by the standard deviation (Mean ± SD). SD: Standard deviation. CUE—Crude Ultrasound Extract; CSE—Crude Soxhlet Extract; ESE—Ethanol Soxhlet Extract. *: not identified.

**Table 5 plants-12-02327-t005:** Toxicity analysis of the test of *Allium cepa* and extracts of leaves of *C. guianensis* obtained by ultrasound and Soxhlet.

Samples	Concentration	ARL ± DP (mm) Mean ± SD	RGI	Effect	GR (%)
Control		43.08 ± 5.8	1		100
CUE	50 μg/mL	58.44 ± 1.29	1.36	GS	135.6
	250 μg/mL	41.11 ± 11.72	0.95	SCE	95.4
	750 μg/mL	19.24 ± 1.77	0.45	GI	44.7
CSE	50 μg/mL	47.40 ± 11.40	1.10	SCE	110
	250 μg/mL	34.83 ± 7.41	0.81	SCE	80.8
	750 μg/mL	20.64 ± 3.06	0.48	GI	47.9
ESE	50 μg/mL	60.29 ± 1.70	1.40	GS	139.9
	250 μg/mL	44.49 ± 11.20	1.03	SCE	103.2
	750 μg/mL	32.34 ± 2.00	0.75	GI	75

Values represent the mean followed by the standard deviation (Mean ± SD), SD—standard deviation. GI—growth inhibition; SCE—same growth effect; GS growth stimulus. SD: Standard deviation. CUE—Crude Ultrasound Extract; CSE—Crude Soxhlet Extract; ESE—Ethanol Soxhlet Extract. ARL—Average root length; RGI—Relative Growth Index; GR—Growth Rate.

**Table 6 plants-12-02327-t006:** Lethal concentrations of 50% (IC_50_) of *C. guianensis* leaf extracts obtained by ultrasound and Soxhlet using the *Artemia salina* test.

Extract	IC_50_ (μg/mL)	Toxicity
CUE	2.318	Non-toxic
CSE	2.308	Non-toxic
ESE	1.478	Non-toxic

CUE—Crude Ultrasound Extract; CSE—Crude Soxhlet Extract; ESE—Ethanol Soxhlet Extract.

**Table 7 plants-12-02327-t007:** Antimicrobial activity of the extracts from the leaves of *C. guianensis* obtained by ultrasound and Soxhlet methods. Mean inhibition halo (in mm) during the action of the extracts against the microorganisms *Staphylococcus aureus* and *Streptococcus mutans*.

Extracts	Concentration (mg)	Microorganisms	
		*S. aureus* Mean ± SD	*S. mutans* Mean ± SD
		Zone of Inibition (mm)	
CUE	50	10.50 ± 0.70 ^b^	7.83 ± 0.49 ^b^
	100	10.44 ± 1.75 ^b^	7.87 ± 0.15 ^b^
	200	12.21 ± 1.23 ^a^	10.31 ± 0.17 ^a^
CSE	50	8.09 ± 0.22 ^c^	4.48 ± 2.89 ^b^
	100	9.61 ± 0.15 ^b^	9.16 ± 0.39 ^a^
	200	10.29 ± 0.04 ^a^	9.99 ± 1.39 ^a^
ESE	50	7.72 ± 2.83 ^b^	8.13 ± 0.15 ^b^
	100	12.57 ± 0.74 ^a^	8.69 ± 0.62 ^b^
	200	13.41 ± 1.76 ^a^	9.06 ± 0.55 ^a^
Chlorhexidine (+)		10.82 ± 0.81 ^a^	12.37 ± 0.61 ^a^
DMSO (−)		0.00 ± 0.00	0.00 ± 0.00

Values followed by the same letter indicate significant similarities in the same column (*p* < 0.05, ANOVA followed by Tukey’s test); values represent the mean followed by the standard deviation (Mean ± SD). (+) positive control; (−) negative control. SD: Standard deviation. CUE—Crude Ultrasound Extract; CSE—Crude Soxhlet Extract; ESE—Ethanol Soxhlet Extract.

## Data Availability

The data presented in this study are available on request from the corresponding author.

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
