# Peer review of "Extracts from the Leaf of Couroupita guianensis (Aubl.): Phytochemical, Toxicological Analysis and Evaluation of Antioxidant and Antimicrobial Activities against Oral Microorganisms"

_plants, 2023, doi:10.3390/plants12122327_

Round 1
Reviewer 1 Report
This is an interesting study wherein the authors studied the antioxidant and antimicrobial properties of Couroupita guianensis against bacteria S. aureus & S mutans. They characterized the leaf extract and found presence of flavonoids, tannins saponins known to possess antioxidant and antimicrobial properties. they also conducted toxicological analysis of the extracts on onion and shrimp. I found the authors lyophilized and freeze-thawed the extracts multiple times during processing. I am curious to know if they did any follow up studies to see if that altered the physico-chemical properties of the extract. I have attached a copy of the manuscript below with my comments

English language is more or less fine.
Author Response
Response to Reviewer 1 Comments
Dear reviewer 1, here are the responses regarding the requested corrections.
Point 1: I found the authors lyophilized and freeze-thawed the extractsmultiple times during processing. I am curious to know if they did any follow up studies to see ifthat altered the physico-chemical properties of the extract. I have attached a copy of themanuscript below with my comments.
Response 1: Initially, the extracts were prepared in soxhlet and ultrasound. Then, the excess solvent used in the extractions was removed in a rotary rotary evaporator. Only then were the samples frozen and then lyophilized. The samples were removed from the freezer and directly lyophilized, so the sample was frozen only once and no changes occurred in the physico-chemical properties of the extracts.
Point 2: I have attached a copy of themanuscript below with my comments
Response 2: All requested corrections in the manuscript were made.
Point 3: English language is more or less fine.
Response 3: The English was corrected by a professional in the area.

Reviewer 2 Report
The study deals with the Phytochemical, Toxicological Analysis and Evaluation of Antioxidant and Antimicrobial Activities Against Oral Microorganisms of the phyto extract from the leaf of Couroupita guianensis (Aubl.). It is a medicinal plant and native to Legal Amazon popularly known in Brazil as abricó de macaco. The study is well prepared and present useful information. Therefore, should be consider for publication in this journal. However, there are some decencies which must be improved.
Use or repetition of similar word in a same sentence should be avoided like check line 24-25.
Line 32 italicize the species name.
Abbreviations must be explained at first use.
Line 50-56 should be cited with relevant studies https://doi.org/10.1016/j.heliyon.2023.e15909.
DOI: 10.36899/JAPS.2022.3.0475, https://doi.org/10.3390/molecules27196728
Line 57-60 are not clear and should be revise.
Add more details of the studied species such as medicinal uses, traditional uses, important phytochemicals in previous studies.
Add novelty of the study and significance of the study in introduction.
Section 4.3 provide complete details of the method
Add study gap and future recommendation in the conclusion.
Some English sentences need improvement such as very long sentences
Author Response
Response to Reviewer 2 Comments
Dear reviewer 2, here are the responses regarding the requested corrections.
Point 1: Use or repetition of similar word in a same sentence should be avoided like check line 24-25.
Response 1: The correction was made and the repeated terms were removed.
Point 2: Line 32 italicize the species name.
Response 2: All corrections have been made to the species name.
Point 3: Line 50-56 should be cited with relevant studies https://doi.org/10.1016/j.heliyon.2023.e15909.
DOI: 10.36899/JAPS.2022.3.0475, https://doi.org/10.3390/molecules27196728
Response 3: References have been added to the text as requested.
Point 4: Line 57-60 are not clear and should be revise. Add more details of the studied species such as medicinal uses, traditional uses, importante phytochemicals in previous studies. Add novelty of the study and significance of the study in introduction.
Response 4: The introduction has been rewritten and the requested points have been emphasized.
Point 5: Section 4.3 provide complete details of the method.
Response 5: The tests used for each qualitative analysis were added, in a summarized form, and the references with the complete methodologies of each one were cited. The methodologies were not described in full due to their size, but if necessary we can add them in full.
Point 6: Add study gap and future recommendation in the conclusion.
Response 6: The conclusion was rewritten as requested.
Point 7: Some English sentences need improvement such as very long sentences
Response 7: The English was corrected by a professional in the area.
Reviewer 3 Report
The manuscript entitled “Extracts From The Leaf Of Couroupita guianensis (Aubl.): Phytochemical, Toxicological Analysis And Evaluation Of Antioxidant And Antimicrobial Activities Against Oral Microorganisms” describes a research conducted in order to discover new natural products which may be used in dentistry, with action against the most important oral diseases, caries and periodontal disease. The results of this study are promising demostrating that C. guianensis has the potential to be a source of great therapeutic interest, with applications in oral health.
The manuscript is clear and well-written and the methods are well planned, well conducted and well described.
The authors were looking to both the accuracy and the reproducibility of all applied methods.
I recommend publication.
Author Response

(The authors gave the same response as above.)

Reviewer 4 Report
Please see the attachment.

Author Response
Response to Reviewer 4 Comments
Dear reviewer 4, here are the responses regarding the requested corrections.
Point 1: Introduction
In general, introduction is poorly written, without significant tone and with a limited number of references. In my opinion, this part should be completely restructured and rewritten.
- Lines 50-56: repetition of the same sentences
- Lines 57-59 and 61-56: repetition of the same sentences
Response 1: The introduction was completely rewritten.
Point 2: What about the ultrasound-based method? This method was adopted from some previous work, all developed in the laboratory? Why did you decide to expose a sample to ultrasound for so long? This can drastically change and destroy the chemical composition of the extract.
Response 2: Extraction with ultrasound has been developed in several works in the laboratory and is in agreement with other works already published in the literature. Several published works use ultrasound as a methodology in the extraction of biomolecules of interest, using long reaction times, all with good results. Therefore, one of the chosen methodologies was this one.
Point 3: Lines 36-37. A brief description of the testing protocols should be included in the manuscript.
Response 3: The qualitative analysis methods used were mentioned briefly and the reference with the complete methodology was mentioned.
Point 4: What about the method HPLC-DAD? Was it developed in the laboratory or was it adopted from the literature? In the first case, the validation protocol for the method should be included, in the second case, you should add references.
Response 4: The method was developed in the laboratory and applied to samples from different plants. A study carried out and published with the method was " Chemical Composition and Photoprotective Potential of Infusion Extract from Casearia sylvestris var. lingua (Cambess.) Eichler Leaves" Orbital: Electron. J. Chem. 2022, 14(2), 89, http://dx.doi.org/10.17807/orbital.v14i2.15578
The reference was included in the manuscript.
Point 5: The comparison of the results with the literature is superficial. The article is very difficult to read, as each sentence is written by itself, without a "red thread" in the text.
Response 5: The discussion were rewritten and the discussion improved as requested.
Point 6: Representative HPLC-DAD chromatogram should be included in the manuscript.
Response 6: The chromatogram was inserted in the manuscript (Figure 1).
Point 7: - How was prepared extract for GC-MS analysis?
Response 7: One hundred mg of the extract was solubilized in 2 ml of water using ultrasound for 1 min and then 2 ml of hexane was added and kept in ultrasound for 2 min. After resting, phase formation occurred, so the hexane fraction was separated from the aqueous fraction. To the aqueous fraction was added 2 ml of hexane, and the process was repeated. After the two extractions, the hexane fractions were dried and suspended in 1000 ml of hexane. For GC-MS analysis, the solution was first filtered through a 0.45 µm Ultrafilter.
Point 8: - Representative GC-MS chromatogram should be included in the manuscript.
Response 8: The data were interpreted and recently we had a problem with the equipment's computer, making it impossible to obtain this chromatogram.
Point 9: Conclusions
- The repetition of the written text from the abstract is not necessary in the conclusion.
Response 9: The conclusion has been rewritten as requested.
Point 9: References
- 22 of the total 54 references are older than 10 years. The reference list should be updated.
Response 9: Old references were removed and references less than 10 years old were added.

Round 2
Reviewer 4 Report
In the case of the HPLC analysis, I would like to see an additional chromatogram of the pure standards. In the case of the GC analysis, I still think that the chromatogram must be shown.
No comments.
Author Response
Dear reviewer 4, here are the responses regarding the requested corrections.
Point 2: - In the case of the HPLC analysis, I would like to see an additional chromatogram of the purestandards. In the case of the GC analysis, I still think that the chromatogram must be shown.
Response 2: We are experiencing problems with our computers regarding stored data. This fact has caused us numerous problems in relation to data obtained until February 2023.
We send the requested chromatograms, which are not in good resolution for publication. We hope that by July 2023 we will be able to recover the analyzes.
The chromatograms are in the attached file.

Round 3
Reviewer 4 Report
Dear authors, thank you for responding to my requests and correcting the manuscript in the spirit of the suggestions.
I still think that the analytics of this paper is not at the highest level. First of all, the HPLC chromatogram of the real sample looks amazingly "clean" and is identical to the chromatogram of the standard components, which you will admit is very unusual. In the case of GC analysis, the sample preparation and the identified components are questionable.
However, the focus of the paper is on other analyses, so I will leave the decision on this paper to the editor.
Finally, please add the appropriate units on all chromatograms.
All the best.
No comments.
Author Response
Response to Reviewer 4 Comments
Dear reviewer 4, here are the responses regarding the requested corrections.
Point 1: Dear authors, thank you for responding to my requests and correcting the manuscript in the spiritof the suggestions.I still think that the analytics of this paper is not at the highest level. First of all, the HPLC chromatogram of the real sample looks amazingly "clean" and is identical to the chromatogramof the standard components, which you will admit is very unusual. In the case of GC analysis,the sample preparation and the identified components are questionable.
However, the focus of the paper is on other analyses, so I will leave the decision on this paper tothe editor.
Finally, please add the appropriate units on all chromatograms.
All the best.
Response 1: Dear Reviewer, As already indicated, we are having problems recovering the original chromatograms. For this reason, we ask that you leave only the chromatogram in figure 1 in the text. If you feel that our GC-MS analysis is not adequate as it was performed, we may remove it from the study. One option we are considering is to request a review period of around 60 days, so that we can redo all the chromatographic analyzes of the extracts. It remains for us to assess whether we managed to recover the original extracts analyzed in order to avoid problems with changes in composition due to the time taken to collect the plant material and storage time. I hope that our technical problems do not affect the final result of the analysis of the article. All other requests were granted.
The manuscript was revised and rewritten according to the reviewers' comments. The changes are marked and the English language has been revised by an expert in the field.
